# Prevention of Cardiovascular Disease and Cancer Mortality by Achieving Healthy Dietary Goals for the Swedish Population: A Macro-Simulation Modelling Study

**DOI:** 10.3390/ijerph16050890

**Published:** 2019-03-12

**Authors:** Sanjib Saha, Jonas Nordstrom, Ulf-G Gerdtham, Irene Mattisson, Peter M Nilsson, Peter Scarborough

**Affiliations:** 1Health Economics Unit, Department of Clinical Science (Malmö), Lund University, Medicon Village, Scheelevägen 2, SE-22381, Lund, Sweden; ulf.gerdtham@med.lu.se; 2Department of Food and Resource Economics, University of Copenhagen, DK-1958 Frederiksberg, Denmark; jonas.nordstrom@agrifood.lu.se; 3Lund University School of Economics and Management, Agrifood Economics Centre, SE-22007 Lund, Sweden; 4Department of Economics, Lund University, SE-22363 Lund, Sweden; 5National Food Agency, SE-75126 Uppsala, Sweden; irene.mattisson@slv.se; 6Department of Emergency Care and Internal Medicine, Skane University Hospital, SE-20502 Malmo, Sweden; peter.nilsson@med.lu.se; 7Department of Clinical Sciences (Malmo), Lund University, SE-20502 Lund, Sweden; 8British Heart Foundation Centre on Population Approaches for Non-Communicable Disease Prevention and NIHR Biomedical Research Centre at Oxford, Nuffield Department of Population Health, University of Oxford, Oxford OX37LF, UK; peter.scarborough@dph.ox.ac.uk

**Keywords:** cardiovascular diseases, diet-related cancers, dietary intake, dietary recommendations

## Abstract

The objective is to estimate the number of deaths attributable to cardiovascular diseases and diet-related cancers that could be prevented or delayed in Sweden if adults adhere to the official dietary recommendations. We used an age-group and sex-specific epidemiological macro-simulation model to estimate preventable deaths due to the discrepancies between actual intake and recommended intake of changes in food components. Data included in the model are a baseline scenario (actual dietary intake), a counterfactual scenario (recommended intake) and age- and sex-specific mortality for cardiovascular and diet-related cancer diseases together compared with the total population risk of a specific year. Monte Carlo analyses with 5000 iterations was performed to produce the 95% uncertainty intervals (UI). The model predicts that 6405 (95% UI: 5086–7086) deaths could be prevented or delayed if the Swedish population could adhere to official dietary recommendations in a year. More deaths would be saved for men than women. The recommendations for fruits and vegetables could have saved 47% of the deaths, followed by fiber intake (32%). For men, fruits and vegetables could have saved more compared to other dietary components, while for women dietary fiber was the prominent factor. Public health policies should consider ensuring healthy eating practices for the Swedish population.

## 1. Introduction

Every eight years Nordic countries revise official nutrient recommendations based on the available information regarding dietary components, nutrient requirements and health. [1]. However, there is a large gap between the recommendations and actual dietary practice in Sweden [2]. Moreover, Sweden has experienced important changes in dietary intake over the past decades [3,4]. For example, information on food supply shows changes, higher supply of energy, lower of milk, higher of meat and vegetables [4]. The most recent dietary survey showed that many participants reported a high intake of foods high in sugar and fat and on average, about 15% of dietary energy was derived from such food sources [3]. Researchers have shown that poorly balanced diets (i.e. high in salt, sugar and saturated fat, but low in fruits, vegetables and fiber) can increase the risk of numerous cancers, cardiovascular diseases and other chronic diseases like type 2 diabetes and obesity [5,6]. Chronic diseases are the most important cause of mortality, morbidity and disability worldwide as well as in Sweden [7]. These diseases also place a substantial burden on national economies and represent one of the major causes of health inequalities [8,9]. For example, obesity and overweight alone have been estimated to account for about 3% of the total direct and indirect costs of all illnesses in Sweden [10].

To reduce the burden of ill health related to poor diet, governmental intervention is justified to encourage healthier food consumption, e.g, via information and taxation. Before that, it is important to identify and quantify the health benefits that can be obtained by switching to a diet conforming to the official dietary recommendations. Such knowledge would provide guidance regarding the prioritization of targeted interventions and where resources should be allocated strategically.

The extent to which the incidences of chronic diseases and/or deaths could be avoided or prevented by modifying the dietary intake of the Swedish population to the recommended intake is still unknown. Moreover, it is also unclear which dietary components could yield the most beneficial effect in health if adhered to by Swedes. One possible reason could be the difficulties to conduct standard epidemiological study designs such as randomized controlled trials for the entire population to compare, for example, increased fruits and vegetables (or dietary fiber) intake vs. reduced salt (or saturated fat) intake between an intervention group and controls [11]. This is either impractical or unethical, for example, in order to estimate the effect of proposed taxes to modify sugar-sweetened beverages intake, a randomized trial would involve randomizing people, shops, or areas to receive increases in food prices.

Simulation modelling is one possible option for synthesizing results from observational studies where experimental studies are difficult [11,12]. A simulation model can combine all the available evidence to estimate a scenario where head to head comparison is not possible. Such models use a collection of mathematical equations to quantify the relationships between interventions and particular outcomes of interests [13]. Furthermore, simulation models have several advantages over clinical trials such as linking intermediate clinical endpoint to final outcomes e.g. changes in blood pressure to hypertensive diseases. Moreover, models can synthesize head-to-head comparisons where no relevant trial exists to compare different dietary components. Last but not the least, models can inform decision making in the absence of reliable data where a particular experiment is not feasible [14].

Therefore, the purpose of this research is, firstly, to quantify health benefits by means of deaths that could be prevented or delayed if the Swedish population could follow dietary recommendations using a simulation model; and, secondly, to identify which dietary components would provide the highest health benefits for the Swedish population.

## 2. Materials and Methods

We used a transparent and well-used macrosimulation model to estimate the cardiovascular and diet-related cancer death toll that could be prevented or delayed for the Swedish population in a year by comparing the actual dietary intake with the recommended dietary intake of the Swedish population.

### 2.1. The Simulation Model

A comparative risk assessment macrosimulation model, PRIME (Preventable Risk Integrated Model) [15] developed in Microsoft Excel has been used. The PRIME model is described in detail elsewhere [15]. In short, the PRIME model simulates the effect of changes in consumption of foods (fruits and vegetables) and nutrients (fat, salt, dietary fiber, energy) through biological risk factors such as blood pressure, serum cholesterol, and overweight/obesity to diet-related mortality from cardiovascular diseases and diet-related cancers. Figure 1 shows the conceptual framework of the model.

To be included in the model, food components have to be recognized as statistically associated with cardiovascular disease outcomes and cancer, or biological risk factors for these diseases. Meta-analyses obtained from randomized controlled trials and prospective cohort studies are used to parameterize changes in nutritional risk factors and mortality as a result of the change in population intake of food items and nutrients [15]. PRIME estimates the differences in mortality in one single year between the baseline scenario (actual dietary intake, in this case) and the counterfactual scenario (recommended dietary intake). The model is based on a number of key assumptions:Combined changes in the risks for individuals are multiplicative. For example, if one extra serving of vegetables reduces the risk of cardiovascular diseases by 12% and reducing salt intake by 1 gram per day reduces the risk by 10%, then both of these behaviour changes jointly reduce the risk of cardio-vascular disease (CVD) death by 20.8% (1 − (1 − 0.12) × (1 − 0.10)).Changes in risk follow a log-linear, dose-response relationship except for obesity, which follows a J-shaped curve. For example, a change in consumptions of fruits and vegetables from 2 to 3 servings has the same effect on relative risk as a change in consumption from 7 to 8 servings. However, an upper threshold has been included, above which there are no additional health benefits. The upper thresholds are based on the range of data collected in the meta-analyses used to parameterise the models. It is unlikely that the effects of different food components are independent and additive. By combining parameters multiplicatively, the PRIME model estimates the overlap in estimated changes in risk of cause-specific mortality as they relate to changes in different dietary components (i.e., the outcome of changing several dietary components simultaneously is less than the sum of its parts and can never exceed 100% risk reduction).

Additionally, the counterfactual scenarios analysed in this paper assume that the population distribution for each of the dietary variables shifts so that the new mean consumption level meets the dietary recommendations, but variance of consumption within the population remains the same.

PRIME has been previously used to answer similar research questions in the UK [16], Canada [17] and France [18].

### 2.2. Actual Intake

Actual average dietary intake of Swedish population is obtained from the most recent Swedish dietary survey “Riksmaten-vuxna 2010–2011” which was conducted by the Swedish National Food Agency (Livsmedelsverket). This cross-sectional survey invited a representative sample of 5003 individuals aged between 18 to 80 years living in Sweden and recruited a total of 1797 participants (36% recruitment rate). Participants reported everything they ate and drank during four consecutive days using a self-reported web-based food questionnaire. The data collection took place between May 2010 and July 2011. This survey was used to estimate population intake of energy, fruits and vegetables, fiber, salt and fats, which includes total fats, saturated fats, poly unsaturated fatty acids (PUFA), monounsaturated fatty acids (MUFA) and cholesterol, stratified by age and sex. The intake of fruits and vegetables were separated, and pulses (e.g. beans, peas, chickpeas, lentils etc.) and roots were included in the vegetables group. In the fruit category, berries were included but not the fruit juice. Details of the data used as model inputs are presented in the Appendix A.

### 2.3. Recommended Intake

We used the nutrient recommendations from the most recent Nordic Nutrition Recommendation, 2012 (NNR) [19]. The NNR provides reference values for the intake of nutrients for individuals living in Nordic countries, which are Denmark, Finland, Iceland, Norway and Sweden. The NNR is based on the results of a thorough evaluation of all relevant research outcomes within the field of nutrition.

The NNR provides the recommendation for energy yielding nutrients as a range. For example, 25%–40% of the energy should come from fat. For this study, we required exact targets for consumption, so we converted these ranges into the best value.

The Swedish National Food Agency (Livsmedelsverket) published the revised version of the National Food-based Dietary Guidelines in 2015 [20]. The guidelines were developed by the National Food Agency. These are also considering the Swedish food culture and the ability of consumers to follow the recommendations. These guidelines combine fruits and vegetables recommendations into one (i.e 500 g/day) but we have separated the intake by 250 g for each component in this study. The actual mean dietary intake and the recommendations are provided in Table 1.

### 2.4. Population Statistics

The model requires age- and sex-specific population mortality for specific diseases for a given year. The mortality data for coronary heart diseases (International Classification of Diseases-10 (ICD-10: I20–25)), stroke (ICD-10: I60–69) and diet-related cancers (ICD-10: C00–14, C16, C23, and C33–34) has been obtained from the Swedish National Board of Health and Welfare (Socialstyrelsen) database. The population statistics have been obtained from the Statistics Sweden database. The latest population and mortality data (from 2016) were used.

### 2.5. Uncertainty Analysis

A Monte Carlo simulation was conducted to estimate the uncertainty intervals around the results. Each of the estimates in the model was allowed to vary according to the distribution reported in the accompanying literature. The 95% uncertainty interval estimates are based on the 2.5th and 97.5th percentile percentiles of results obtained from 5000 iterations of the model.

## 3. Results

The reported intake of fruits and vegetables was low for both Swedish men and women compared to the recommended intake (Table 1). Women consumed more fruits and vegetables than men. However, in terms of fiber, men had higher intake than women although these intakes were far from the recommended intake. The total fat and saturated fat intake were higher than recommended, whereas MUFA and PUFA intake was lower than recommended. Men consumed more salt than women and both consumptions were higher than the recommended intake level.

The model estimates that 6405 (95% UI: 5086–7086) deaths could be prevented or delayed in a year if the Swedish population could follow the dietary recommendations (Table 2), which is 14.4% of the total deaths related to the diseases included in the model for that specific year. Of these, more deaths would be prevented or delayed among men than women. For men, modifying the dietary intake to meet the fruits and vegetables recommendations would have the highest potential to prevent deaths (53%) followed by dietary fiber (20%) and salt (18%). For women, dietary fiber intake is associated with the highest benefit, i.e. 50% deaths would have been avoided followed by fruits and vegetables (42%) and fats (9%).

Most of the deaths that could be prevented or delayed by improving the dietary intake of Swedish population would be related to coronary heart diseases (Table 3), followed by stroke. The scenario is the same for both men and women. In terms of cancer, only lung cancer and colorectal cancers were influenced by simulated dietary changes to some degree. Around 24% death from cardiovascular diseases and 10% from cancer could be prevented or delayed if the Swedish population adhere to the dietary guidelines.

## 4. Discussion

In this simulation model study, we show that a considerable number of deaths could be prevented or delayed if the Swedish population could follow dietary recommendations. We find that most of the lives could be saved by the changes attributable to an increase in fruits and vegetables consumption followed by fiber intake.

It is interesting to note that men would have gained higher health benefits from increased intake of fruits and vegetables whereas women would have gained most from increased fiber intake. The reason is that men consumed fewer fruit and vegetable servings than women whereas the fiber intake was low in women compared to men (Table 1). This raises a question mark, as fruits and vegetables contain a substantial amount of fiber. However, in the simulation model the fruits, vegetables and fiber intake were separated. Moreover, men have in general a higher calorie intake than women, and the bulk of calories for men comes from grains. Grains represent a high source of fiber, especially whole grains [21]. The benefits from recommended salt intake is the third highest according to the simulation model. However, it should be noted that the intake of salt estimation was underestimated with the method used in the “Riksmaten-vuxna 2010–2011”. This means that the effect we estimate probably is also underestimated. Another simulation study showed that salt reduction will have significant impact on stroke, ischemic heart disease and mortality in Sweden over 20-years period [22]. However, in this study, they assumed 20% discretionary use of salt in addition to salt intake from food records.

Changes in intake of fats and fatty acids have much fewer benefits than fruits and vegetables and fiber. Estimates applied in this study depend on both the strength of the association between dietary factors and health outcomes. The actual intake of fat for the Swedish population lies very close to the recommended intake, and is similar to what has been reported from Canada [17] although Swedish population have high intake of saturated fat. Epidemiological studies suggested that diets rich in MUFA and PUFA are associated with low mortality [23,24].

The results from this study can be compared with studies that modeled the health impact of achieving dietary recommendations in Canada [17] and UK [16], using the earlier version of the PRIME model. The UK study suggested that 46% of the deaths averted or delayed could be attributed to meeting the fruits and vegetables recommendations [16], with a further 23% from achieving the salt recommendation. For Canada, it was 72% and 10% for fruits and vegetables and salt, respectively [17]. For the Swedish data, the figures are close to the UK study for the fruits and vegetables intake and the Canadian study for the salt intake. The reason might be the difference between recommended fruits and vegetables consumption in Sweden, Canada and the UK. The recommendation for the UK is five servings per day [25] (equivalent to 400 grams) and at least seven servings (depending on sex and age) in Canada [17], whereas in Sweden the recommendation is 500 grams per day excluding fruit juice [19]. Moreover, the Swedish population has a so called Nordic diet heritage which is protective for cardiovascular diseases and certain cancers as a recent study indicates [26]. The Nordic food culture is different from food cultures from the UK and Canada. Since the food culture, as well as the recommendations are country-specific it justifies studying the health benefits for each country. Moreover, using the same health simulation model provides the extra benefit of cross-country comparisons.

The impact of achieving each of the recommendations was estimated in isolation and combined together which is a strength of this study. Furthermore, the dietary survey was the actual intake of the Swedish population for four consecutive days compared to other studies where the data collected from food purchase survey in the UK [16] and New Zealand [27] but only by 24-hour recall survey for a single day in Canada [17]. Moreover, the vital statistics and population data were obtained from credible sources and Sweden has a long reputation of good epidemiological data due to unique personal identity number and validated national registers [28]. The Cause of Death Registers and population registers are well maintained with a high coverage rate [29].

PRIME is a transparent model for which all the risk equations related to change in diet and mortality comes from high-quality meta-analyses [15]. Moreover, the model has been used several times in different countries, for example in the UK [25,30,31] and Ireland [32], New Zealand [27], France [18], Canada [17] and Denmark [33]. The estimates of relative risks that have been used to parameterize the model were taken from results of published meta-analyses. However, not all of the studies included in the meta-analyses adjusted their results for each of the dietary factors or biological risk factors that are included in the model. For example, the relative risks for cardiovascular diseases per increase in body mass index might not have adjusted for different fat intakes and vice versa [15]. Similarly, the effect of fruits and vegetables on cardiovascular diseases is likely to be partially mediated by dietary fiber, which is not accounted for in the model. Therefore, it is likely that some degree of double counting affects the model, and that (on these grounds) the deaths prevented or delayed might be overestimated.

The model provides a cross-sectional analysis that is how much deaths will be prevented in a particular year and does not provide estimates for a longer time period, for example deaths prevented in five- or 10-year intervals such as the DYNAMO-HIA model [22] which provides estimate over 20-years. Moreover, the model provides the health gains in terms of deaths prevented or delayed. The quantification of the health gain in terms of health or health-related Quality-Adjusted Life Years (QALY) gained would have been useful as this could capture both the quantity and quality of life years gained as well as the productivity loss due to early mortality. The societal consequences of death of an individual older than 65 is different from that of a 45-year old person. Furthermore, the model needs to update with risk equation so that not only mortality but also the prevention of non-fatal disease events can be captured e.g. in the same way as done in some renowned diabetes model [34]. It might be helpful to perform an economic evaluation of several interventions targeted to increase healthy dietary habits.

Additionally, we assume that the health benefits will be gained in the same year if people follow the NNR. It would however, take years (e.g. effects of salt reduction on cardiovascular disease) [35] or even decades (e.g. effects of dietary fiber on cancer), for the full health gain to be realized. Furthermore, the dietary survey was conducted in 2011 and we estimate the health benefits for the year 2016. We assume that the dietary pattern has not changed so much since the last dietary survey which can be seen as a conservative. One obvious limitation was that the dietary survey had a low participation rate (36%) and the participation rate was lower in men than women [3]. The level of education was somewhat higher among participants than in non-participants, which might indicate that the participants were more health selected and more health conscious than those who did not participate. Thus, the survey may provided a too positive pictures of food habits of the Swedish population. Therefore, the model estimation of death averted or delayed might be an underestimation.

Although we have accounted for the uncertainty in the risk estimation by providing 95% uncertainty intervals, there is also some variation i.e. uncertainties in the survey or the conversion of the food consumed into nutrients which we do not account for. Whilst the PRIME model accounts for competing risks by combining relative risks multiplicatively, it does not account for any interactions between risk factors.

This simulation model study has potential for future research. We identified that 14.4% of death could be prevented in a year by adhering to a healthy diet, for which fruits, vegetables and dietary fiber are the prominent risk-reducing constituents. The next step would be to analyze how the policy makers can intervene to increase the consumptions of fruits and vegetables and dietary fiber, e.g. via changed relative prices between healthy and less healthy food products by using taxes or subsidies [36,37], or by workplace interventions (free fruit, healthy meals in the canteen) [38,39]. For Sweden, it was found that the recommended dietary fiber intake can be reached via a subsidy on grain products, but also that the subsidy resulted in an increased intake of fat, salt and sugar, which suggest that both subsidies and taxes should be used [40,41]. For Denmark, a study showed that the saturated fat tax in Denmark (16 Danish Krona per kilogram of saturated fat) have reduced the intake of saturated fat (−4%) in the population and also resulted in increased consumption of vegetables, fruits and fiber [33]. However, from a policy perspective it also important to study the effectiveness of the policy instrument that is used to improve the dietary intake. For example, the effectiveness of an excise tax vs. an ad valorem or excise tax, and the effectiveness of taxing/subsidizing various food products. Besides effectiveness, a policy has to be cost-effective in terms of the size of effects obtained for the money spent. A simulation model where different tax and subsidies rates can be tested in terms of their cost-effectiveness might help decision makers to take an informed decision.

## 5. Conclusions

In conclusion, this study shows that by modifying dietary intake, a considerable number of deaths could be prevented or delayed in Sweden. Thus, policy makers should take necessary steps to modify the dietary intake of the Swedish population.

## Figures and Tables

**Figure 1 ijerph-16-00890-f001:**
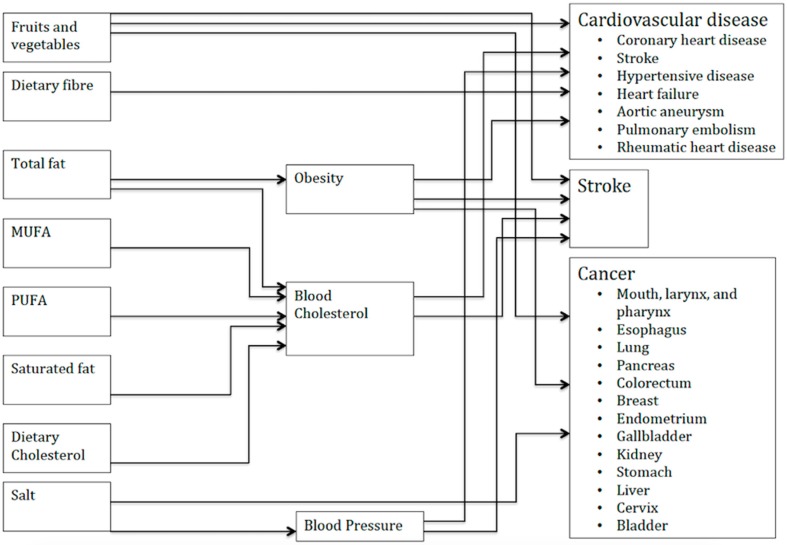
Conceptual framework for the Preventable Risk Integrated Model (PRIME).

**Table 1 ijerph-16-00890-t001:** Mean dietary component intake versus recommended intake for Swedish men and women.

Food/Nutrient	Recommended Intake	Actual Mean Intake (SE)	
		Men (n = 792)	Women (n = 1005)
Fruits (g/day) *	250.0	105.0 (3.97)	147.0 (3.53)
Vegetables (g/day) *	250.0	169.0 (3.69)	182.0 (3.09)
Fibre (g/day)	30.00	21.30 (0.29)	18.80 (0.22)
Salt (g/day)	6.00	8.84 (0.10)	6.78 (0.063)
Total fat (%E)	40.00	34.0 (0.21)	34.40 (0.20)
Saturated fat (%E)	9.00	13.0 (0.11)	13.10 (0.10)
MUFA (%E) *	20.00	12.80 (0.09)	12.90 (0.09)
PUFA (%E)	10.00	5.5 (0.067)	5.7 (0.06)
Cholesterol (mg/day) *	300	320 (5.15)	263 (3.9)

* Significantly different at *p* = 0.05 between men and women. Abbreviations: SE, Standard error of mean; %E, percentage of total energy; MUFA, Monounsaturated fatty acids; PUFA, Polyunsaturated fatty acids. Note: 1. Actual mean intakes are based on data from Riksmaten-vuxna 2010–2011. 2. Recommended intakes are based on Nordic Nutrition Recommendation 2012. 3. Recommended intake for fruits and vegetables together in 500 g/day excluding fruit juice. The amount was divided equally for fruits and vegetables. 4. Recommended intakes for fatty acids are provided as range and a single value is used.

**Table 2 ijerph-16-00890-t002:** Estimated number of total deaths averted or delayed by specific dietary changes according to guidelines in a year in Sweden.

	Men (Mean, 95% UI)	Women (Mean, 95% UI)	Total (Mean, 95% UI)
Fruits and vegetables	1905 (1262–2152)	1073 (811–1420)	3013 (2080–3566)
Fiber	718 (512–1275)	1285 (656–1577)	2025 (1197–2792)
Fats	623 (471–792)	245 (224–487)	969 (709–1274)
Salt	666 (335–1175)	180 (63–237)	1057 (391–1423)
All dietary guidelines combined	3626 (2994–4175)	2553 (2030–2980)	6405 (5086–7086)

UI, Uncertainty interval. Note: Due to the stochastic nature of the model, the total figure might not be the same for adding up male and female together.

**Table 3 ijerph-16-00890-t003:** Estimated numbers of deaths averted or delayed by cause if Swedish men and women adhered to dietary guidelines.

Causes of Death	Men (Mean, 95% UI)	Women (Mean, 95% UI)	Total (Mean, 95% UI)
Cardiovascular Diseases			
Coronary heart disease	2532 (1913–2775)	1623 (1266–2028)	4077 (4529–6462)
Stroke	551 (407–777)	609 (302–822)	1219 (729–1548)
Heart failure	72 (37–139)	30 (9–36)	147 (46–175)
Aortic aneurysm	27 (12–47)	4 (2–6)	49 (14–54)
Pulmonary embolism	6 (2–11)	1 (0–3)	10 (2–14)
Rheumatic Heart disease	1 (0–3)	0 (0–1)	2 (1–4)
Hypertensive disease	120 (59–200)	54 (18–65)	233 (77–267)
Actual mortality	12,206	11,093	23,299
Cancer			
Colo-rectal	76 (10–241)	90 (32–178)	261 (96–363)
Lung	240 (103–381)	142 (45–174)	407 (141–564)
Actual mortality	3251	3141	6392

UI, Uncertainty interval. Note: 1. Due to the stochastic nature of the model, the total figure might not be the same for adding up male and female together. 2. The actual mortality for these diseases in 2016 is obtained from the Swedish National Board of Health and Welfare (Socialstyrelsen) database.

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
