# Peer review of "Prevention of Cardiovascular Disease and Cancer Mortality by Achieving Healthy Dietary Goals for the Swedish Population: A Macro-Simulation Modelling Study"

_ijerph, 2019, doi:10.3390/ijerph16050890_

Round 1

Reviewer 1 Report

Thank you to the authors for the opportunity to review their work. 

In general, the authors used a basic structure to complete their analysis, essentially applying previous/other's methods to their population of interest. While this isn't necessarily original, the authors did this deliberately as the methodology was validated in previous work (e.g. assessing the impact of changes in the UK). The value of the work is how they demonstrate the strong potential to save lives (hence my 'High' rating for 'Significance of Content' and 'Interest to the readers').

Here are my specific comments for their consideration:

Major:

I don't understand exactly how the author's applied PRIME. (a) Did the estimates allow for competing risks (meaning how do multiple changes for the same person affect the lives saved count) or did they assess the effect of population changes for each dietary changes individually. Does the model say how many lives are saved if you increase those meeting the Fruits recommendation and change nothing else or is it saying this is the impact of increasing those meeting the Fruits recommendation after accounting for an increase in those meeting the recommendations in the other categories? Please explain how the total lives saved in Table 2 ('All dietary guidelines combined') relates to the individual guidelines in the table, especially as the sum of the individual guidelines is more than the total (so, the combined lives counts allows for competing risks?) (b) Was the counterfactual scenario just that all in the population now met the recommended amounts? That is, do you go from percent failing to meet the recommended amounts at baseline to 0 percent failing to meet (if so, is it valuable to look at the impact of different increases in percents meeting the guidelines)? Or is there some range of change like everyone increased their intake some percent of the intake (like the mean grams of fruits increased 50 grams)? If it is the first, then Table 1 should include the baseline percent meeting the recommended intake? If it is the second, then the methods need elaboration on the simulation scenarios (what they were and why were they chosen?). I know the supplemental materials try to make this clear, but at least this reviewer wasn't clear on what exactly was the difference in the countefactual (c) Why are they called 'credible' sets? I am used to seeing this in a Bayesian context, but I don't see how PRIME is Bayesian. The PRIME authors call them uncertainty intervals.

Do the authors have any concerns about the large gap in time between the mortality counts     (2016) and the intake variables (2011), especially as the generalization for the results is to the effect in a single year (that is, the simulation was not based on intake and mortality values obtained from the same year)? The authors should at least mention this point and explain why it is not a concern. Also, the authors claim the intake data is representative (line 124) but in line 272 discuss concerns with the survey. 

Minor Comments:

A small number of typos. For example 'years' in line 41, shows in line 45, the sentence starting with 'Simulation model' on line 72 is not clear, and ', etc.' in line 132, and 6405 not 6.405 in line 176 (and the values in the CI). 

The authors say 'fewer benefits' (line 216) but I want to make sure they mean this as fewer lives saved at the population level. This is not clear from the 'benefits' language.

Is it possible to relate specific guidelines to the specific mortality causes?

Line 232. There is a point made about the protective properties of the Nordic food culture. However, would it be interesting to test this by seeing if it (guidelines based on it) really does save more lives than other guidelines? 

What software did you use for the PRIME simulation?

Author Response

Reviewer 1

Thank you to the authors for the opportunity to review their work. 

In general, the authors used a basic structure to complete their analysis, essentially applying previous/other's methods to their population of interest. While this isn't necessarily original, the authors did this deliberately as the methodology was validated in previous work (e.g. assessing the impact of changes in the UK). The value of the work is how they demonstrate the strong potential to save lives (hence my 'High' rating for 'Significance of Content' and 'Interest to the readers').

Here are my specific comments for their consideration:

Major:

Authors’ Response:

We would like to thank the reviewer for good and valuable comments. We presented our responses and actions for each of the comments below.

Comments # 1:

I don't understand exactly how the author's applied PRIME. (a) Did the estimates allow for competing risks (meaning how do multiple changes for the same person affect the lives saved count) or did they assess the effect of population changes for each dietary changes individually. Does the model say how many lives are saved if you increase those meeting the Fruits recommendation and change nothing else or is it saying this is the impact of increasing those meeting the Fruits recommendation after accounting for an increase in those meeting the recommendations in the other categories? Please explain how the total lives saved in Table 2 ('All dietary guidelines combined') relates to the individual guidelines in the table, especially as the sum of the individual guidelines is more than the total (so, the combined lives counts allows for competing risks?)

Authors’ Response:

The PRIME model accounts for competing risks by combining relative risks multiplicatively. This is already explained (with an example) in lines 107-110 of the manuscript. For reporting the results by individual risk factor, we model the impact of achieving those recommendations in isolation – for this reason, the sum of the individual risk factors is greater than the total number of lives saved. The PRIME model is unable to account for interactions between risk factors (e.g. if increasing fruit and vegetable consumption provides more health benefit for high salt consumers than low salt consumers). We have added a line to the limitations to highlight this.

Authors’ Action:

Changes in the manuscript

In the method section

It is unlikely that the effects of different food components are independent and additive. By combining parameters multiplicatively, the PRIME model estimates the overlap in estimated changes in risk of cause-specific mortality as they relate to changes in different dietary components (i.e., the outcome of changing several dietary components simultaneously is less than the sum of its parts and can never exceed 100% risk reduction).

In the discussion section

Whilst the PRIME model accounts for competing risks by combining relative risks multiplicatively, it does not account for any interactions between risk factors.

Comments # 2:

(b) Was the counterfactual scenario just that all in the population now met the recommended amounts? That is, do you go from percent failing to meet the recommended amounts at baseline to 0 percent failing to meet (if so, is it valuable to look at the impact of different increases in percents meeting the guidelines)? Or is there some range of change like everyone increased their intake some percent of the intake (like the mean grams of fruits increased 50 grams)? If it is the first, then Table 1 should include the baseline percent meeting the recommended intake? If it is the second, then the methods need elaboration on the simulation scenarios (what they were and why were they chosen?). I know the supplemental materials try to make this clear, but at least this reviewer wasn't clear on what exactly was the difference in the counterfactual

Authors’ Response:

The counterfactuals are based on changing dietary variables that are continuous (e.g. fruit consumption (g/d)), rather than binary exposures (meet recommendations for fruit (yes/no)). Therefore, we have a distribution of each variable within the population, which we use as our baseline. For the counterfactuals, we shift the distribution so that the new mean level of consumption matches the recommendation, but the variance in the population remains the same as in the baseline. This is equivalent to everyone in the population making the same change to their diet. This means that approximately 50% of the population will still not meet the recommendations in the counterfactual scenario, but this is appropriate since population-level targets (such as dietary recommendations) are monitored by tracking population mean consumption levels.

Authors’ Action:

Changes in the manuscript

Additionally, the counterfactual scenarios analyzed in this paper assume that the population distribution for each of the dietary variables shifts so that the new mean consumption level meets the dietary recommendations, but variance of consumption within the population remains the same.

Comments # 3:

 (c) Why are they called 'credible' sets? I am used to seeing this in a Bayesian context, but I don't see how PRIME is Bayesian. The PRIME authors call them uncertainty intervals.

Authors’ Response:

This is based on the Monte Carlo analysis with 2.5th and 97.5th percentile. We changed it to uncertainty intervals.

Authors’ Action:

Changed throughout the manuscript.

Comments # 4:

Do the authors have any concerns about the large gap in time between the mortality counts (2016) and the intake variables (2011), especially as the generalization for the results is to the effect in a single year (that is, the simulation was not based on intake and mortality values obtained from the same year)? The authors should at least mention this point and explain why it is not a concern. Also, the authors claim the intake data is representative (line 124) but in line 272 discuss concerns with the survey.

Authors’ Response:

1.     Since we do not have any updated dietary intake survey, we used the latest one and assume that the dietary pattern is still the same. However, we have discussed this issue in the manuscript.

2.     A representative sample was invited but the participation rate was low (36%). In that sense, the participated sample was not representative.

Authors’ Action:

Changes in the manuscript

In the discussion section

Furthermore, the dietary survey was conducted in 2011 and we estimated the health benefits for the year 2016. We assume that the dietary pattern has not changed so much since the last dietary survey which can be seen as a conservative.

In the method section

This cross-sectional survey invited a representative sample of 5003 individuals between 18 to 80 years living in Sweden and recruited a total of 1797 participants (36% recruitment rate)

Minor Comments:

Comments # 5:

A small number of typos. For example 'years' in line 41, shows in line 45, the sentence starting with 'Simulation model' on line 72 is not clear, and ', etc.' in line 132, and 6405 not 6.405 in line 176 (and the values in the CI). 

Authors’ Response:

Thanks. The typos have been corrected.

Authors’ Action:

Changed in the manuscript accordingly.

Comments # 6:

The authors say 'fewer benefits' (line 216) but I want to make sure they mean this as fewer lives saved at the population level. This is not clear from the 'benefits' language.

Is it possible to relate specific guidelines to the specific mortality causes?

Authors’ Response:

We have rewritten the sentence.

Authors’ Action:

Changes in the manuscript

Changes in intake of fats and fatty acids averted less death than fruits and vegetables and fiber.

Comments # 7:

Line 232. There is a point made about the protective properties of the Nordic food culture.

However, would it be interesting to test this by seeing if it (guidelines based on it) really does save more lives than other guidelines?

Authors’ Response:

We agree that this is an interesting research question. However, we think that this should be done in a separate/new study.

Authors’ Action:

None

Comments # 8:

What software did you use for the PRIME simulation?

Authors’ Response:

The PRIME was developed in MS excel. We reported it in the manuscript.

Authors’ Action:

A comparative risk assessment macrosimulation model, PRIME (Preventable Risk Integrated ModEl) [15]  developed in Microsoft Excel has been used.

Reviewer 2 Report

Thank you for the opportunity to review this report. The authors had found  that by using this PRIME model, 6,405 (95% confidence interval: 5,086-7,086) deaths could be prevented or delayed. Recommended fruits and vegetables could save 47% of the deaths, followed by fiber intake. I feel this report is well designed and rational and posess novelty. Still, I have some comments that may help the readers to understand the report more efficiently.

The authors obtained dietary information with usage of response to a national questionnaire survey targeted to 1,979 participants. It is always difficult to conduct a dietary survey throughout the whole population. Therefore, survey on sample is rational.  However, how do the authors consider about the sample bias? How will it effect the result? This should be explained in the text.

The authors made usage of the Nordic Nutrition Recommendation. On which theory is this recommendation based on?

In accordance with the previous question, when comparing the actual intake with the Recommendation, I wonder how many deaths will occur when an individual take follow this recommendation intake. No explanation was seen in the reference on PRIME model (Ref #15). Explanation on this would help the readers to better understand.

Why is there no consideration about total energy? Total energy is associated with various diseases.

Can fruit and vegetables be separated in analysis? As fiber will be a common factor of these diets, I think it is better to separate when applying the result from this paper to the actual policy.

Author Response

Thank you for the opportunity to review this report. The authors had found that by using this PRIME model, 6,405 (95% confidence interval: 5,086-7,086) deaths could be prevented or delayed. Recommended fruits and vegetables could save 47% of the deaths, followed by fiber intake. I feel this report is well designed and rational and posess novelty. Still, I have some comments that may help the readers to understand the report more efficiently.

Authors’ Response:

We would like to thank the reviewer for good and valuable comments. We presented our responses and actions for each of the comments below.

Comments # 1:

The authors obtained dietary information with usage of response to a national questionnaire survey targeted to 1,979 participants. It is always difficult to conduct a dietary survey throughout the whole population. Therefore, survey on sample is rational.  However, how do the authors consider about the sample bias? How will it effect the result? This should be explained in the text.

Authors’ Response:

We have already discussed the limitation of the survey. Furthermore, we have included a paragraph in the discussion section explaining the possible effect of sample bias in on the result.

Authors’ Action:

Changes in the manuscript

One obvious limitation was that the dietary survey had a low participation rate (36%) and the participation rate was lower in men than women [3]. The level of education was somewhat higher among participants than in non-participants, which might indicate that the participants were more health selected and more health conscious than those who did not participate. Thus, the survey may provide a too positive pictures of food habits of the Swedish population. Therefore, the model estimation of death averted or delayed might be an underestimation.

Comments # 2:

The authors made usage of the Nordic Nutrition Recommendation. On which theory is this recommendation based on?

Authors’ Response:

The Nordic Nutrition Recommendation was based on existing scientific evidence to set a dietary reference value for the Nordic population that will ensure optimal nutrition and helps prevent lifestyle-related diseases. Experts conducted systematic reviews for different topics and nutrients. A working group including experts from all Nordic counties were responsible for setting the actual reference values based on the systematic reviews and published in the “Nordic Nutrition Recommendations 2012” report.

Authors’ Action:

None

Comments # 3:

In accordance with the previous question, when comparing the actual intake with the Recommendation, I wonder how many deaths will occur when an individual take follow this recommendation intake. No explanation was seen in the reference on PRIME model (Ref #15). Explanation on this would help the readers to better understand.

Authors’ Response:

In the model, we have to include age and sex-specific actual mortality for a particular year from 24 diseases (presented in detail in the technical report, Ref #15).  In table 3, we have presented the actual mortality from the cardiovascular diseases and cancers of the Swedish population in the year 2016.  We have also presented this in the result section.

Authors’ Action:

Changes in the manuscript

In the result section

Table 3 and

Around 24% death from cardiovascular diseases and 10% from cancer could be prevented or delayed if the Swedish population adhere to the dietary guidelines.

Comments # 4:

Why is there no consideration about total energy? Total energy is associated with various diseases.

Authors’ Response:

The model also considers age and sex specific energy intake for both male and female population.

Authors’ Action:

None

Comments # 5:

Can fruit and vegetables be separated in analysis? As fiber will be a common factor of these diets, I think it is better to separate when applying the result from this paper to the actual policy.

Authors’ Response:

We have presented the findings separately for fruit and vegetables and fiber also. A considerable portion of fiber also comes from grains. Furthermore, we have discussed this issue in line 265.

Authors’ Action:

None